# Unsupervised Learning of Graph Hierarchical Abstractions with Differentiable Coarsening and Optimal Transport

## Abstract

Hierarchical abstractions are a methodology for solving large-scale graph problems in various disciplines. Coarsening is one such approach: it generates a pyramid of graphs whereby the one in the next level is a structural summary of the prior one. With a long history in scientific computing, many coarsening strategies were developed based on mathematically driven heuristics. Recently, resurgent interests exist in deep learning to design hierarchical methods learnable through differentiable parameterization. These approaches are paired with downstream tasks for supervised learning. In practice, however, supervised signals (e.g., labels) are scarce and are often laborious and expensive to obtain. In this work, we propose an unsupervised approach, coined OTCOARSENING, with the use of optimal transport. Both the coarsening matrix and the transport cost matrix are parameterized, so that an optimal coarsening strategy can be learned and tailored for a given set of graphs. We demonstrate that the proposed approach produces meaningful coarse graphs and yields competitive performance compared with supervised methods for graph classification and regression.

## 1 Introduction

A proliferation of graph neural networks emerged as competitive alternatives to graph kernels for various graph applications; e.g., theorem proving (Wang et al., 2017) and chemoinformatics (Jin et al., 2017; Fout et al., 2017; Schütt et al., 2017). These models learn sophisticated feature representations of a graph and its constituents (i.e., nodes and edges) through layers of feature transformation. Among them, a broad array of work is convolutional, extending convolution filters in the spatial domain to the spectral domain or to local neighborhoods (Bruna et al., 2014; Henaff et al., 2015; Duvenaud et al., 2015; Defferrard et al., 2016; Kipf & Welling, 2017; Hamilton et al., 2017; Chen et al., 2018; Veličković et al., 2018; Ying et al., 2018a; Liao et al., 2019; Xu et al., 2019b); whereas a few others are recurrent, which treat the representation of a graph node as the state of a dynamical system, being recurrently updated (Scarselli et al., 2009; Li et al., 2016; Gilmer et al., 2017; Jin et al., 2017). Several convolution architectures (Xu et al., 2019b; Morris et al., 2019) are connected to the Weisfeiler–Lehman (WL) graph isomorphism test because of the resemblance in iterative node (re)labeling. They are stated to be as expressive as WL in isomorphism tests, rendering strong competitors to WL graph kernels (Shervashidze et al., 2011) inspired by the same procedure.

An image analog of graph neural networks is convolutional neural networks, whose key components are convolution and pooling. The pooling operation reduces the spatial dimensions of an image and forms a hierarchical abstraction through successive downsampling. For graphs, a similar hierarchical abstraction is particularly important for maintaining the structural information and deriving a faithful feature representation. A challenge, however, is that unlike image pixels that are spatially regular, graph nodes are irregularly connected and hence pooling is less straightforward.

Several graph neural networks perform pooling in a hierarchical manner. Bruna et al. (2014) build a multiresolution hierarchy of the graph with agglomerative clustering, based on $\epsilon$-covering. Defferrard et al. (2016) and Fey et al. (2018) employ Graclus that successively coarsens a graph based on the heavy-edge matching heuristic. Simonovsky & Komodakis (2017) construct the hierarchy through a combined use of spectral polarity and Kron reduction. These neural networks build the

graph hierarchy in the preprocessing step, which defines in advance how pooling is performed given a graph. No learnable parameters are attached. In fact, most practical coarsening methods to date are built on mathematical heuristics; how they affect the structure and properties of the graph is less understood (Loukas & Vandergheynst, 2018).

Recently, hierarchical abstractions as a learnable neural network module surfaced in the literature of graph representation learning. Representative approaches include DIFFPOOL (Ying et al., 2018b), GRAPH U-NET (Gao & Ji, 2019), and SAGPOOL (Lee et al., 2019). In the first approach, a soft clustering of nodes is parameterized and learned. The next graph in the hierarchy is thus a complete graph of the clusters. In the second approach, the top nodes according to some parameterized ordering are selected and the induced subgraph becomes the next graph in the hierarchy. The third approach is similar to the second one, except that the ordering is computed through self-attention. All approaches treat the learnable hierarchy as part of the neural network (in conjunction with a predictive model), which is trained with a downstream task in a (semi-)supervised manner.

In this work, we propose an unsupervised approach, called OTCOARSENING, that produces a hierarchical abstraction of a graph independent of downstream tasks. Therein, node features for the graphs in the hierarchy are derived simultaneously, so that they can be used for different tasks through training separate downstream predictive models. OTCOARSENING consists of two ingredients: a parameterized graph coarsening strategy in the algebraic multigrid (AMG) style; and an optimal transport that minimizes the structural transportation between two consecutive graphs in the hierarchy. The "OT" part of the name comes from Optimal Transport. We show that this unsupervised approach produces meaningful coarse graphs that are structure preserving; and that the learned representations perform competitively with supervised approaches in graph classification and regression.

The contribution of this work is threefold. First, for unsupervised learning we introduce a new technique based on hierarchical abstraction through minimizing discrepancy along the hierarchy. Second, key to a successful hierarchical abstraction is the coarsening strategy. We develop one motivated by AMG and empirically show that the resulting coarse graphs qualitatively preserve the graph structure. Third, we demonstrate that the proposed technique, combining coarsening and unsupervised learning, performs comparably with supervised approaches but is advantageous in practice facing label scarcity.

## 2 RELATED WORK

Hierarchical (a.k.a. multilevel or multiscale) methods are behind the solutions of a variety of problems, particularly for graphs. Therein, coarsening approaches are being constantly developed and applied. Two active areas are graph partitioning and clustering. The former is often used in parallel processing, circuit design, and solutions of linear systems, among many others. The latter appears in descriptive data analysis. Several representative developments are discussed here. It is not intended to be a full account of the overwhelming literature and long history.

Many of the graph hierarchical approaches consist of a coarsening and an uncoarsening phase. The coarsening phase successively reduces the size of a given graph, so that an easy solution can be obtained for the smallest one. Then, the small solution is lifted back to the original graph through successive refinement in the reverse coarsening order. For coarsening, a class of approaches applies heave-edge matching heuristics (Hendrickson & Leland, 1995; Karypis & Kumar, 1998; Dhillon et al., 2007). In the conceptual level, nodes connected by a heavily weighted edge are grouped into a node in the coarse graph, so that the edge is protected from partitioning. The use of matching heuristics was not much analyzed until recently. Loukas and coauthors show that for certain graphs, the principal eigenvalues and eigenspaces of the coarsened and the original graph Laplacians are close under randomized matching (Loukas & Vandergheynst, 2018; Loukas, 2019). On the other hand, in the uncoarsening phase, refinement can be done in several ways. One approach is Kernighan-Lin refinement (Kernighan & Lin, 1970), which is commonly applied in spectral partitioning and spectral clustering methods (Shi & Malik, 2000; Luxburg, 2007), whether or not done in a multilevel fashion. Another approach uses kernel $k$-means, as in Dhillon et al. (2007).

Another class of coarsening approaches selects a subset of nodes from the original graph. Call them coarse nodes; they form the node set of the coarse graph. Other nodes are aggregated with weights to the coarse nodes in certain ways, which, simultaneously define the edges in the coarse graph.

Many of these methods were developed akin to algebraic multigrid (AMG) (Ruge & Stüben, 1987), as also is this work. In AMG, the set $C$ of coarse nodes is initialized as empty. Then, each node in the complement of $C$ is investigated in some order; if its coupling to the current $C$ is sufficiently weak, the node is moved to $C$. The coupling may be defined based on edge weights (Kushnir et al., 2006), diffusion distances (Livne & Brandt, 2012), or algebraic distances (Ron et al., 2011; Chen & Safro, 2011; Safro et al., 2014). The aggregation weights are defined accordingly. In this work, the selection of the coarse nodes and the aggregation weights are parameterized and learned instead. Besides the AMG style, the dominant eigenvector of the graph Laplacian has also been used for selecting coarse nodes (Shuman et al., 2015), who however use a combination of Kron reduction (Dörfler & Bullo, 2013) and graph sparsification to define the edges of the coarse graph.

Hierarchical graph representation is emerging in graph deep learning. Representative approaches include DIFFPOOL (Ying et al., 2018b), GRAPH U-NET (Gao & Ji, 2019), and SAGPOOL (Lee et al., 2019). Cast in the above setting, DIFFPOOL is more similar to the first class of coarsening approaches, whereas GRAPH U-NET and SAGPOOL more similar to the latter. All methods are supervised, as opposed to ours.

Our work is additionally drawn upon optimal transport, a tool recently used for defining similarity of graphs (Vayer et al., 2019; Xu et al., 2019a). In the referenced work, Gromov–Wasserstein distances are developed that incorporate both node features and graph structures. Moreover, a transportation distance from the graph to its subgraph is developed by Garg & Jaakkola (2019). Our approach is based on a relatively simpler Wasserstein distance, whose calculation admits an iterative procedure more friendly to neural network parameterization.

## 3 METHOD

In this section, we present the proposed method OTCOARSENING, beginning with the two main ingredients: coarsening and optimal transport, followed by a summary of the computational steps in training and the use of the results for downstream tasks.

### 3.1 AMG-STYLE COARSENING

The first ingredient coarsens a graph $G$ into a smaller one $G_c$. For a differentiable parameterization, an operator will need be defined that transforms the corresponding graph adjacency matrix $A \in \mathbb{R}^{n \times n}$ into $A_c \in \mathbb{R}^{m \times m}$, where $n$ and $m$ are the number of nodes of $G$ and $G_c$ respectively, with $m < n$. We motivate the definition by algebraic multigrid (Ruge & Stüben, 1987), because of the hierarchical connection and a graph-theoretic interpretation. AMG also happened to be referenced as a potential candidate for pooling in some graph neural network architectures (Bruna et al., 2014; Defferrard et al., 2016).

#### 3.1.1 BACKGROUND ON ALGEBRAIC MULTIGRID

AMG belongs to the family of multigrid methods (Briggs et al., 2000) for solving large, sparse linear systems of the form $Ax = b$, where $A$ is the given sparse matrix, $b$ is the right-hand vector, and $x$ is the unknown vector to be solved for. For simplicity, we assume throughout that $A$ is symmetric. The simplest algorithm, two-grid V-cycle, consists of the following steps: (i) Approximately solve the system with an inexpensive iterative method and obtain an approximate solution $x'$. Let $r = b - Ax'$ be the residual vector. (ii) Find a tall matrix $S \in \mathbb{R}^{n \times m}$ and solve the smaller residual system $(S^T A S)y = S^T r$ for the shorter unknown vector $y$. (iii) Now we have a better approximate solution $x'' = x' + Sy$ to the original system. Repeat the above steps until the residual is sufficiently small.

In practice, it is unlikely that the residual system $(S^T A S)y = S^T r$ in the second step, though smaller, can be solved exactly, if the original system $Ax = b$ cannot be. Hence, one naturally appeals to recursion. That is, one solves $(S^T A S)y = S^T r$ only approximately, obtains the residual, constructs a further smaller residual system, and proceeds recursively, until when a sufficiently small residual system can be solved exactly and inexpensively.

The matrix of the residual system, $S^T A S$, is called the Galerkin coarse-grid operator. One may show that step (ii), if solved exactly, minimizes the energy norm of the error $x - x''$ over all possible corrections from the range of the matrix $S$. Decades of efforts on AMG discover practical definitions

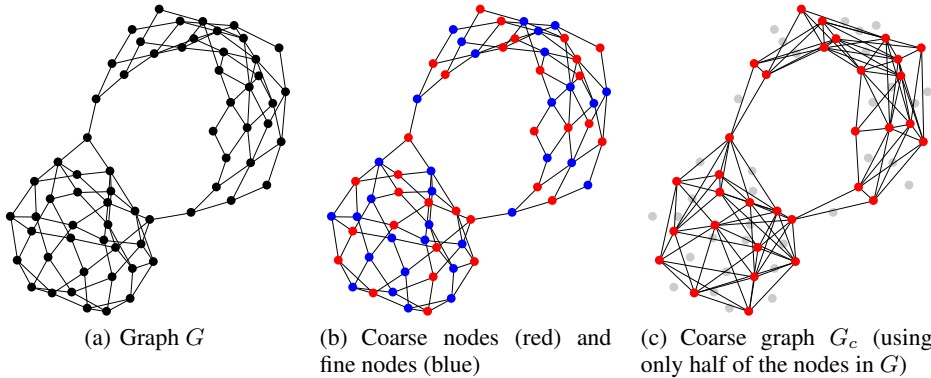

(a) Graph $G$

(b) Coarse nodes (red) and fine nodes (blue)

(c) Coarse graph $G_c$ (using only half of the nodes in $G$)

Figure 1: Example graph and coarsening.

of $S$ that both is economic to construct/apply and encourages fast convergence. We depart from these efforts and define/parameterize an $S$ that best suites graph representation learning.

### 3.1.2 COARSENING FRAMEWORK

Following the above motivation, we settle with the coarsening framework

$$A_c = S^T A S, \tag{1}$$

where $S$ is named the *coarsening matrix*. For parameterization, we might have treated $S$ as a parameter matrix, but it requires a fixed size to be learnable and hence it can only be applied to graphs of the same size. This restriction both is unnatural in practice and destroys permutation invariance of the nodes. In what follows, we discuss the properties of $S$ from a graph theoretic view, which leads to a natural parameterization.

### 3.1.3 PROPERTIES OF $S$

Let $V$ be the node set of the graph $G$. AMG partitions $V$ into two disjoint subsets $C$ and $F$, whose elements are called *coarse nodes* and *fine nodes*, respectively. See Figure 1(b). For coarsening, $C$ becomes the node set of the coarse graph and the nodes in $F$ are eliminated.

The rows of the coarsening matrix $S$ correspond to the nodes in $V$ and columns to nodes in $C$. This notion is consistent with definition (1), because the rows and columns of $A_c$ correspond to the coarse nodes. It also distinguishes from DIFFPOOL (Ying et al., 2018b), which although defines the next graph by the same equation (1), does not use the nodes in the original graph as those of the smaller graph.

If $S$ is dense, so is $A_c$. Then, the graphs in the coarsening hierarchy are all complete graphs, which are less desirable. Hence, we would like $S$ to be sparse. Assuming so, one sees that each column of $S$ plays the role of aggregation. Consider the matrix vector product $y = S^T x$. The value of the $j$-th node in the coarse graph, $y_j$, is an aggregation of the values of the original nodes $x_1, \ldots, x_n$, where the aggregation weights come from the $j$-th column of $S$. For convenience, we define $\chi(j)$ to be the set of nonzero locations of this column and call it the *aggregation set* of the coarse node $j$. The following result characterizes the existence of an edge in the coarse graph. Its proof is a simple exercise; see the appendix.

**Theorem 1.** *There is an edge connecting two nodes $j$ and $j'$ in the coarse graph if and only if there is an edge connecting the two aggregation sets $\chi(j)$ and $\chi(j')$ in the original graph.*

Hence, in order to encourage sparsity of the coarse graph, many of the aggregation set pairs should not be connected by an edge. One principled approach to ensuring so, is to restrict the aggregation set to contain at most direct neighbors and the node itself. The following corollary is straightforward. We say that the *distance* of two nodes is the number of edges in the shortest path connecting them.

**Corollary 2.** *If each aggregation set contains at most direct neighbors and the node itself, then there is an edge connecting two nodes in the coarse graph only if the distance between them in the original graph is at most 3.*

Consequently, in what follows we will let $S$ have the same sparsity structure as the corresponding part of $A + I$. The identity matrix is used to introduce self loops. An illustration of the resulting coarse graph is given in Figure 1(c), with self loops omitted.

### 3.1.4 PARAMETERIZATION OF $S$

With the graph-theoretic interpretation of $S$, we now parameterize it. The strategy consists of the following computational steps. First, select coarse nodes in a differentiable manner, so that the sparsity structure of $S$ is determined. Then, compute the nonzero elements of $S$.

The selection of coarse nodes may be done in several ways, such as the top-k approach that orders nodes by projecting their feature vectors along a learnable direction (see, e.g., Cangea et al. (2018); Gao & Ji (2019)). This approach, however, leverages only node features but not the graph information. To leverage both, we apply one graph convolution

$$\alpha = \text{sigmoid}(\widehat{A}XW_\alpha) \tag{2}$$

to compute a vector $\alpha \in \mathbb{R}^{n \times 1}$ that weighs all nodes (Lee et al., 2019). Here, $\widehat{A} \in \mathbb{R}^{n \times n}$ is the normalized graph adjacency matrix defined in graph convolutional networks (Kipf & Welling, 2017), $X \in \mathbb{R}^{n \times d}$ is the node feature matrix, and $W_\alpha \in \mathbb{R}^{d \times 1}$ is a parameter vector. The weighting necessitates using sigmoid (or other invertible functions) rather than ReLU as the activation function. Naturally, one may explore using more than one graph convolution layer to enrich model expressiveness. See implementation details in the appendix.

For a coarsening into $m$ nodes, we pick the top $m$ values of $\alpha$ and list them in the sorted order. Denote by $\alpha_s \in \mathbb{R}^{m \times 1}$ such a vector, where the subscript $s$ means sorted and picked. We similarly denote by $\widehat{A}_s \in \mathbb{R}^{n \times m}$ the column-sorted and picked version of $\widehat{A}$.

We let $S$ be an overlay of the graph adjacency matrix with the node weights $\alpha_s$. Specifically, define

$$S = \ell_1\text{-row-normalize}[\widehat{A}_s \odot (\mathbf{1}\alpha_s^T)], \tag{3}$$

where $\mathbf{1}$ means a column vector of all ones.

There are several reasons why $S$ is so defined. First, $S$ carries the nonzero structure of $\widehat{A}_s$, which, following Corollary 2, renders more likely a sparse coarse graph. Second, the use of the normalized adjacency matrix introduces self loops, which ensure that an edge in the coarse graph exists if the distance is no more than three, rather than exactly three (which is too restrictive). Third, because both $\widehat{A}_s$ and $\alpha_s$ are nonnegative, the row normalization ensures that the total edge weight of the graph is preserved after coarsening. To see this, note that $\mathbf{1}^T A_c \mathbf{1} = \mathbf{1}^T S^T A S \mathbf{1} = \mathbf{1}^T A \mathbf{1}$.

## 3.2 OPTIMAL TRANSPORT

The second ingredient of the proposed OTCOARSENING uses optimal transport for unsupervised learning. Optimal transport (Peyré & Cuturi, 2019) is a framework that defines the distance of two probability measures through optimizing over all possible joint distributions of them. If the two measures lie on the same metric space and if the infinitesimal mass transportation cost is a distance metric, then optimal transport is the same as the Wasserstein-1 distance. In our setting, we extend this framework for defining the distance of the original graph $G$ and its coarsened version $G_c$. Then, the distance constitutes the coarsening loss, from which model parameters are learned in an unsupervised manner.

### 3.2.1 OPTIMAL TRANSPORT DISTANCE

To extend the definition of optimal transport of two probability measures to that of two graphs $G$ and $G_c$, we treat the node features from each graph as atoms of an empirical measure. The coarse node features result from graph neural network mappings, carrying information of both the initial node

features and the graph structure. Hence, the empirical measure based on node features characterizes the graph and leads to a natural definition of graph distance.

Specifically, let $M$ be a matrix whose element $M_{ij}$ denotes the transport cost from a node $i$ in $G$ to a node $j$ in $G_c$. We define the distance of two graphs as

$$W_\gamma(G, G_c) := \min_{P \in U(a,b)} \langle P, M \rangle - \gamma E(P), \tag{4}$$

where $P$, a matrix of the same size as $M$, denotes the joint probability distribution constrained to the space $U(a, b) := \{P \in \mathbb{R}_+^{n \times m} \mid P\mathbf{1} = a, \ P^T\mathbf{1} = b\}$ characterized by marginals $a$ and $b$; $E$ is the entropic regularization $E(P) := -\sum_{i,j} P_{ij}(\log P_{ij} - 1)$; and $\gamma > 0$ is the regularization magnitude. The first term $\langle P, M \rangle$ is the usual definition of the transportation cost between two discrete measures. Because we treat them as empirical measures, each of $a$ and $b$ has constant elements that sum to unity, respectively. As is well known, the optimum of $\langle P, M \rangle$ is unstable and the cost of obtaining it through linear programming is high. A popular remedy is the entropic regularization (Wilson, 1969), $E(P)$, which brings in an additional benefit that the optimization (4) admits a computational procedure that is friendly to parameterizations inside $M$.

Through a simple argument of Lagrange multipliers, it is known that the optimal $P_\gamma$ that solves (4) exists and is unique, in the form

$$P_\gamma = \mathrm{diag}(u)K\,\mathrm{diag}(v),$$

where $u$ and $v$ are certain positive vectors of matching dimensions and $K = \exp(-M/\gamma)$ with the exponential being element-wise. The solution $P_\gamma$ may be computationally obtained by using Sinkhorn's algorithm (Sinkhorn, 1964): Starting with any positive vector $v^0$, iterate

$$\text{for } i = 0, 1, 2, \ldots \text{ until convergence, } u^{i+1} = a \oslash (Kv^i) \text{ and } v^{i+1} = b \oslash (K^T u^{i+1}). \tag{5}$$

Because the solution $P_\gamma$ is part of the loss function to be optimized, we cannot iterate indefinitely. Hence, we instead define a computational solution $P_\gamma^k$ by iterating only a finite number $k$ times:

$$P_\gamma^k := \mathrm{diag}(u^k)K\,\mathrm{diag}(v^k). \tag{6}$$

Accordingly, we arrive at the $k$-step optimal transport distance

$$W_\gamma^k(G, G_c) := \langle P_\gamma^k, M \rangle - \gamma E(P_\gamma^k). \tag{7}$$

The distance (7) is the sample loss for training.

### 3.2.2 Parameterization of $M$

With the distance defined, it remains to specify the transport cost matrix $M$. As discussed earlier, we model $M_{ij}$ as the distance between the feature vector of node $i$ from $G$ and that of node $j$ from $G_c$. This approach on the one hand is consistent with the Wasserstein distance and on the other hand, carries both node feature and graph structure information.

Denote by $\mathrm{GNN}(A, X)$ a generic graph neural network architecture that takes the graph adjacency matrix $A$ and node feature matrix $X$ as input and produces as output a transformed feature matrix. We produce the feature matrix $X_c$ of the coarse graph through the following encoder-decoder-like architecture:

$$Z = \mathrm{GNN}(A, X), \quad Z_c = S^T Z, \quad X_c = \mathrm{GNN}(A_c, Z_c). \tag{8}$$

The encoder produces an embedding matrix $Z_c$ of the coarse graph through a combination of GNN transformation and aggregation $S^T$, whereas the decoder maps $Z_c$ to the original feature space so that the resulting $X_c$ lies in the same metric space as $X$. Then, the transport cost, or the metric distance, $M_{ij}$ is the $p$-th power of the Euclidean distance of the two feature vectors:

$$M_{ij} = \|X(i,:) - X_c(j,:)\|_2^p. \tag{9}$$

In this case, the optimal transport distance is the $p$-th root of the Wasserstein-$p$ distance. The power $p$ is normally set as one or two.

---

**Algorithm 1** Unsupervised training: forward pass

---
1: **for** each coarsening level **do**
2:     Compute coarsening matrix $S$ according to (2) and (3)
3:     Perform coarsening and obtain $A_c$ and $X_c$ according to (1) and (8)
4:     Obtain also node embeddings $Z_c$ from the computation (8)
5:     Compute transport cost matrix $M$ according to (9)
6:     Compute $k$-step joint probability estimate $\hat{P}_\gamma^k$ according to (5) and (6)
7:     Compute current-level loss ($k$-step optimal transport distance) $W_\gamma^k(G, G_c)$ according to (7)
8:     Set $G \leftarrow G_c$, $A \leftarrow A_c$, and $X \leftarrow X_c$
9: **end for**
10: Sum the loss for all coarsening levels as the sample loss

---

### 3.3 TRAINING AND DOWNSTREAM USE

With the technical ingredients developed in the preceding subsections, we summarize the computational steps into Algorithm 1, which is self explanatory.

After training, for each graph $G$ we obtain a coarsening sequence and the corresponding node embedding matrices $Z_c$ for each graph in the sequence. These node embeddings may be used for a downstream task. Take graph classification as an example. For each node embedding matrix, we perform a global pooling (e.g., a concatenation of max pooling and mean pooling) across the nodes and obtain a summary vector. We then concatenate the summary vectors for all coarsening levels to form the feature vector of the graph. A multilayer perceptron is then built to predict the graph label.

## 4 EXPERIMENTS

In this section, we conduct a comprehensive set of experiments to evaluate the performance of the proposed method OTCOARSENING. Through experimentation, we aim at answering the following questions. (i) As an unsupervised hierarchical method, how well does it perform on a downstream task, compared with supervised approaches and unsupervised non-hierarchical approaches? (ii) How does the choice of hyperparamters specific to this method affect the performance? (iii) In a multi-task setting, how well does it perform compared with supervised models trained separately for each task? (iv) Do the coarse graphs carry the structural information of the original graphs (i.e., are they meaningful)?

### 4.1 SETUP

We perform experiments with the following data sets: PROTEINS, MUTAG, NCI109, IMDB-BINARY (IMDB-B for short), IMDB-MULTI (IMDB-M for short), and DD. They are popularly used benchmarks publicly available from Kersting et al. (2016). Except IMDB-B and IMDB-M which are derived from social networks, the rest of the data sets all come from the bioinformatics domain. Information of the data sets is summarized in Table 3 in the appendix. The downstream task is graph classification.

We gauge the performance of OTCOARSENING with several supervised approaches. They include the plain GCN (Kipf & Welling, 2017) followed by a gloabl mean pooling, as well as five more sophisticated pooling methods: SORTPOOL (Zhang et al., 2018), which retains the top-k nodes for fixed-size convolution; DIFFPOOL (Ying et al., 2018b), which applies soft clustering; SET2SET (Vinyals et al., 2015), which is used together with GRAPHSAGE (Hamilton et al., 2017) as a pooling baseline in Ying et al. (2018b); GPOOL (Cangea et al., 2018; Gao & Ji, 2019), which retains the top-k nodes for graph coarsening, as is used by GRAPH U-NET; and SAGPOOL (Lee et al., 2019), which applies self-attention to compute the top-k nodes. Among them, DIFFPOOL, GPOOL, and SAGPOOL are hierarchical methods, similar to ours.

Additionally, we employ a simple unsupervised baseline. Named GRAPHAE-UNSUPV, this baseline is a graph autoencoder that does not perform coarsening, but rather, applies GCN twice to respectively encode the node features and decode for reconstruction. The encoder serves the same purpose as that of the plain GCN and the decoder is needed for training without supervised signals.

We implement the proposed method and the graph autoencoder by using the PyTorch Geometric library, which is shipped with off-the-shelf implementation of all other compared methods. We refer the readers to the appendix for implementation and experimentation details. The code is available at https://github.com/anonymousOPT/OTCoarsening.

## 4.2 GRAPH CLASSIFICATION

Graph classification accuracies are reported in Table 1. OTCOARSENING outperforms the compared methods in five out of six data sets: PROTEINS, MUTAG, IMDB-B, IMDB-M, and DD. Moreover, it improves significantly the accuracy on DD. Interestingly, on these data sets the supervised runner up is always DIFFPOOL, outperforming the subsequently proposed GPOOL and SAGPOOL. On the other hand, these two methods perform the best on the other data set NCI109, with SAGPOOL taking the first place. On NCI109, OTCOARSENING performs on par with the lower end of the compared methods. It appears low-performing, possibly because of the lack of useful node features that play an important role in the optimal transport distance. Based on these observations, we may conclude that hierarchical methods indeed are promising for handling graph structured data. Moreover, as an unsupervised method, the proposed OTCOARSENING performs competitively with strong supervised approaches. In fact, even for the simple unsupervised baseline GRAPHAE-UNSUPV, it outperforms DIFFPOOL on PROTEINS, MUTAG, and DD. This observation indicates that unsupervised approaches are quite competitive, paving the way for possible uses in other applications and tasks.

Table 1: Graph classification accuracy.

| Method | PROTEINS | MUTAG | NCI109 | IMDB-B | IMDB-M | DD |
|---|---|---|---|---|---|---|
| GCN | 0.723 | 0.734 | 0.696 | 0.713 | 0.505 | 0.718 |
| SET2SET | 0.734 | 0.746 | 0.703 | 0.729 | 0.497 | 0.708 |
| SORTPOOL | 0.735 | 0.801 | 0.691 | 0.716 | 0.499 | 0.737 |
| DIFFPOOL | 0.742 | 0.845 | 0.717 | 0.743 | 0.503 | 0.739 |
| GPOOL | 0.722 | 0.762 | 0.724 | 0.730 | 0.495 | 0.715 |
| SAGPOOL | 0.733 | 0.786 | **0.731** | 0.722 | 0.504 | 0.720 |
| GRAPHAE-UNSUPV | 0.743 | 0.846 | 0.664 | 0.724 | 0.499 | 0.765 |
| OTCOARSENING | **0.749** | **0.856** | 0.685 | **0.746** | **0.509** | **0.772** |

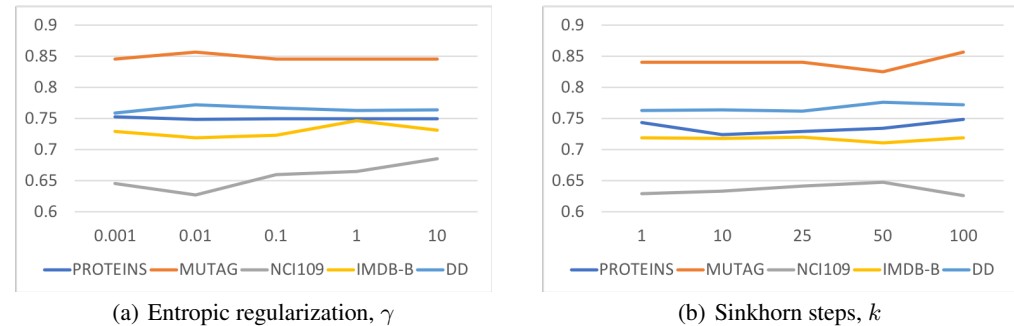

(a) Entropic regularization, $\gamma$    (b) Sinkhorn steps, $k$

Figure 2: Classification accuracy as parameters vary.

## 4.3 SENSITIVITY ANALYSIS

OTCOARSENING introduces a few parameters owing to the computational nature of optimal transport: (a) the entropic regularization strength $\gamma$; and (b) the number of Sinkhorn steps, $k$. In Figure 2, we perform a sensitivity analysis and investigate the change of classification accuracy as these parameters vary. One sees that most of the curves are relatively flat, except the case of $\gamma$ on NCI109. This observation indicates that the proposed method is relatively robust to the parameters of optimal transport. The curious case of NCI109 inherits the weak performance observed in the preceding subsection, possibly caused by the lack of informative input features.

## 4.4 Multi-Task Learning

We further investigate the value of unsupervised graph representation through the lens of multi-task learning. We compare three scenarios: (A) a single representation trained without knowledge of the downstream tasks (method: OTCoarsening); (B) a single representation trained jointly with all downstream tasks (methods: GCN, Set2Set, SortPool, DiffPool, gPool, and SAGPool, all suffixed with "-joint"); and (C) different representations trained separately with each task (method: DiffPool-sep).

The data set is QM7b (Wu et al., 2018), which consists of 14 regression targets. Following Gilmer et al. (2017), we standardize each target to mean 0 and standard deviation 1; we also use MSE as the training loss but test with MAE. Table 2 reports the MAE and timing results.

Table 2: Multi-task regression error and training time (in seconds).

| | Method | MAE | Time | | Method | MAE | Time |
|---|---|---|---|---|---|---|---|
| (A) | OTCoarsening | 0.6609 | 2622 | (B) | SortPool-joint | 2.4408 | 2652 |
| (C) | DiffPool-sep | 0.1714 | 15520 | (B) | DiffPool-joint | 2.4231 | 1100 |
| (B) | GCN-joint | 2.4225 | 2122 | (B) | gPool-joint | 2.4200 | 2117 |
| (B) | Set2Set-joint | 2.4256 | 2657 | (B) | SAGPool-joint | 2.4221 | 1874 |

One sees from Table 2 that in terms of regression error, single unsupervised representation (A) significantly outperforms single supervised representations (B), whilst being inferior to separate supervised representations (C). Each scenario outperforms another at the cost of longer training time. It is expected that (C) is much slower than other scenarios, because it trains 14 models whereas others only 1. The timings for (B) are comparable with that of (A). The timing variation is caused by several factors, including the architecture difference and dense-versus-sparse implementation. DiffPool is implemented with dense matrices, which may be faster compared with other methods that treat the graph adjacency matrix sparse, when the graphs are small.

## 4.5 Qualitative Study

As discussed in Section 2, coarsening approaches may be categorized in two classes: clustering based and node-selection based. Methods in the former class (e.g., DiffPool) coarsen a graph through clustering similar nodes. In graph representation learning, similarity of nodes is measured by not only their graph distance but also the closeness of their feature vectors. Hence, two distant nodes bear a risk of being clustered together if their input features are similar. In this case, the graph structure is destroyed.

On the other hand, methods in the latter class (e.g., Graph U-Net and SAGPool) use nodes in the original graph as coarse nodes. If the coarse nodes are connected based on only their graph distance but not feature vectors, the graph structure is more likely to be preserved. Such is the case for OTCoarsening, where only nodes within a 3-hop neighborhood are connected. Such is also the case for Graph U-Net and SAGPool, where the neighborhood is even more restricted (e.g., only 1-hop neighborhood). However, if two coarse nodes are connected only when there is an edge in the original graph, these approaches bear another risk of resulting in disconnected coarse graphs.

Theoretical analysis is beyond scope. Hence, we conduct a qualitative study and visually inspect the coarsening results. In Figure 3, we show a few graphs from the data set MUTAG, placing the coarsening sequence of OTCoarsening on the left and that of SAGPool on the right for comparison. The hollow nodes are selected as coarse nodes.

For the graph on the top row, OTCoarsening selects nodes across the consecutive rings in the first-level coarsening, whereas SAGPool selects the ring in the middle. For the graph in the middle row, both OTCoarsening and SAGPool select the periphery of the honeycomb for the first-level coarsening, but differ in the second level in that one selects again the periphery but the other selects the heart. For the graph at the bottom row, OTCoarsening preserves the butterfly topology through coarsening but the result of SAGPool is hard to comprehend. This qualitative study corroborates that the coarse graphs produced by OTCoarsening are meaningful.

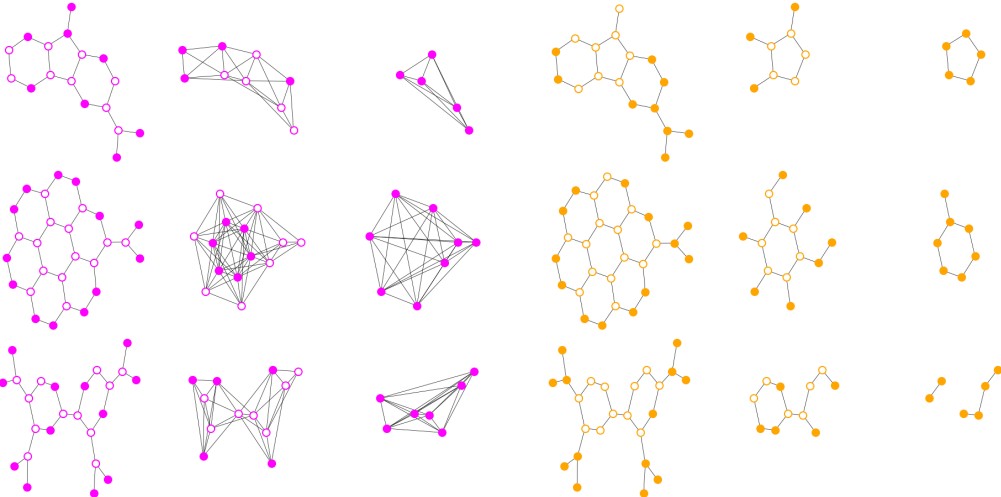

Figure 3: Coarsening sequence for graphs from MUTAG. Left (magenta): OTCOARSENING. Right (orange): SAGPOOL. Hollow nodes are coarse nodes.

## 5    CONCLUDING REMARKS

Coarsening is a common approach for solving large-scale graph problems in various scientific disciplines. It generates a sequence of successively smaller graphs, each one being a structural summary of the prior one, so that the challenging solution with the original graph may be obtained through interpolation, starting from the easy solution with the coarsest graph and interpolating back with refinement, following the reverse order of coarsening. This idea is adopted in the AMG method for solving large, sparse linear systems of equations. When applied to machine learning, the same idea may be explored for learning a hierarchical abstraction of graphs, which are a challenging analog of images that are comprised of regularly connected pixels, because node connections in graphs are generally irregular. How one effectively aggregates nearby nodes and coarsens the graph motivates the present work.

Whereas a plethora of coarsening methods were proposed in the past and are used today, these methods either do not have a learning component, or have parameters that need be learned with a downstream task. In this work, we present OTCOARSENING, which is an unsupervised approach. It follows the concepts of AMG but learns the selection of the coarse nodes and the coarsening matrix through the use of optimal transport. We demonstrate its successful use in graph classification and regression tasks and show that the coarse graphs preserve the structure of the original one. We envision that the proposed idea and architecture may be adopted in many other graph learning scenarios and downstream tasks, such as graph autoencoders, structure learning, and generative modeling.

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

## A  PROOFS

*Proof of Theorem 1.* We say that the sum of two numbers is *structurally nonzero* if at least one of the numbers is nonzero, even if they sum algebraically to zero (e.g., when one number is the opposite number of the other). Structural nonzero of an element in the adjacency matrix is the necessary and sufficient condition for the existence of the corresponding edge in the graph.

Recall that $A_c = S^T A S$. For two coarse nodes $j$ and $j'$, one sees that the element $A_c(j, j')$ is structurally nonzero if and only if the submatrix $A(\chi(j), \chi(j'))$ is nonempty. In other words, $j$ and $j'$ are connected by an edge in the coarse graph $G_c$ if and only if there exists an edge connecting $\chi(j)$ and $\chi(j')$ in the original graph $G$. Note that such an edge may be a self loop. □

*Proof of Corollary 2.* If there is an edge connecting $j$ and $j'$ in the coarse graph, then according to Theorem 1, there is an edge connecting $i \in \chi(j)$ and $i' \in \chi(j')$ in the original graph, for some nodes $i$ and $i'$. Then by the assumption that the elements of $\chi(j)$ are either $j$ or $j$'s direct neighbors and similarly for $\chi(j')$, we know that $j$ and $j'$ are connected by the path $\{j, i, i', j'\}$, which means that the distance between $j$ and $j'$ is at most 3. □

## B  DATA SET DETAILS

See Table 3 for a summary of the classification data sets used in this paper.

Table 3: Data sets.

|                    | PROTEINS | MUTAG | NCI109 | IMDB-B | IMDB-M | DD     |
|--------------------|----------|-------|--------|--------|--------|--------|
| # Graphs           | 1,113    | 188   | 4,127  | 1,000  | 1,500  | 1,178  |
| # Classes          | 2        | 2     | 2      | 2      | 3      | 2      |
| Average # nodes    | 39.06    | 17.93 | 29.68  | 19.77  | 13.00  | 284.32 |
| Average node degree| 3.73     | 2.21  | 2.17   | 9.76   | 10.14  | 5.03   |

## C  IMPLEMENTATION DETAILS

The weighting vector $\alpha$ (cf. (2)) used for coarse node selection is computed by using 1-layer GCN with activation function $\text{sigmoid} \circ \text{square}$. That is, $\alpha = \text{sigmoid}((\widehat{A}XW_\alpha)^2)$.

The GNNs in (8) for computing the coarse node embeddings $Z_c$ and coarse node features $X_c$ are 1-layer GCNs.

The power $p$ in Wasserstein-$p$ (cf. (9)) is fixed as 2.

## D    EXPERIMENTATION DETAILS

We evaluate all methods using 10-fold cross validation.

For training, we use the Adam optimizer (Kingma & Ba, 2014) with a tuned initial learning rate and a fixed decay rate $0.5$ for every $50$ epochs.

We perform unsupervised training for a maximum of 200 epochs and choose the model at the best validation loss. Afterward, we feed the learned representations into a 2-layer MLP and evaluate the graph classification performance.

We use grid search to tune hyperparameters: the learning rate is from $\{0.01, 0.001\}$; and the number of coarsening levels is from $\{1, 2, 3\}$ for the propoed method and $\{2, 3, 4\}$ for the compared methods. The coarsening ratio is set to $0.5$ for all methods.

