# OpenReview forum: "Unsupervised Learning of Graph Hierarchical Abstractions with Differentiable Coarsening and Optimal Transport"
_ICLR.cc/2020/Conference — Reject_

### Official Review · AnonReviewer2 · 2019-10-16
**Official Blind Review #2**

**Rating:** 3

**Review:**

This paper proposes a method to summarize a given graph based on the algebraic multigrid and optimal transport, which can be further used for the downstream ML tasks such as graph classification.
Although the problem of graph summarization is a relevant task, there are a number of unclear points in this paper listed below:

- In Section 3.1, the coarsening method has been proposed, which is said to be achieved by finding S such that A_C = S^T A S.
    However, A_C is usually not binary for S \in R^{n x m}, hence how to get the coarse graph G_C from A_C is not clear. Please carefully explain this point.
- In the proposed method, coarse nodes should be selected beforehand. Is there any guideline of how to choose them?
- In Section 3.2, optimal transport is introduced and the distance between G and G_C is measured via entropic optimal transport in Equation (4) or (7).
    However, in Equation (4), a and b should come from the input G and G_C , and it is not clearly explained how to obtain them from the input.
    Moreover, how to use the distance between G and G_C in the proposed coarsening method is also not clear. It seems that it is not used in Algorithm 1.
- I do not understand why the k-step optimal transport distance is needed. Since it converges to the global optimum as k becomes large, it is usually enough to set k to be large enough.
- In experiments, how is the proposed method used for graph classification?
    Since the proposed method is for generating coarse graphs in an unsupervised manner, graph classification cannot be directly performed by itself.
- In addition to the above issue, to assess the effectiveness of the the proposed method, the following experiment is recommended:
    Fix some classifier and compare performance of graph classification for the original graphs and for the coarse graphs.
- In the qualitative study in Section 4.4, while the authors discuss coarse nodes, they are just an input from the user and results are arbitrary. Hence such discussion is not informative.

Minor comments:
- What is "X" in Equation (2)?
- I recommend to write domain for matrices when they used at the first time.


**Experience Assessment:**

I have published one or two papers in this area.

**Review Assessment: Checking Correctness Of Derivations And Theory:**

I assessed the sensibility of the derivations and theory.

**Review Assessment: Checking Correctness Of Experiments:**

I carefully checked the experiments.

**Review Assessment: Thoroughness In Paper Reading:**

I read the paper at least twice and used my best judgement in assessing the paper.

---

> ### Author Response · Authors · 2019-11-13
> **Response to Official Blind Review #2**
>
> Thank you very much for the detailed questions. In what follows we answer them one by one and hope the replies help you reassess the value of our work. Please do not hesitate to ask more questions should they arise.
>
> - A_c indeed is generally not binary. We treat both A and A_c as *weighted* adjacency matrices. By the design of S, the nonzero elements of A_c are all positive and they qualify as edge weights.
>
> - Coarse nodes are not selected beforehand. Rather, they are selected through a parameterized ranking process (see eqn (2)). The parameters are obtained by optimizing a loss function.
>
> - a and b are empirical measures in the optimal transport setting. As we explain in the text below eqn (4), “each of a and b has constant elements that sum to unity, respectively.”
>
> The optimal transport distance is the loss function that we minimize. This is not to be confused with the fact that the distance itself is the result of a separate minimization problem. The k-step distance is like a closed-form formula solution of this separate minimization problem.
>
> - The use of k-step is to make it a deterministic differentiable function. One may set k however large one wants, but it must be a specific number. In practice, because of the chain rule of differentiation, making k very large incurs a computational burden.
>
> - Graph classification is performed through training a separate predictive model by taking the graph representation as input. As stated in the paper, “For each node embedding matrix, we perform a global pooling (e.g., a concatenation of max pooling and mean pooling) across the nodes and obtain a summary vector. We then concatenate the summary vectors for all coarsening levels to form the feature vector of the graph. A multilayer perceptron is then built to predict the graph label.”
>
> - As indicated by the above item, we use the information of the original graph and all subsequent coarse graphs when performing graph classification. That is, we do not separate these two pieces of information.
>
> - As responded in an earlier item, the selection of the coarse nodes is learned rather than hand-picked. The purpose of Section 4.4 is to show that the learned result is meaningful qualitatively.
>
> - X in eqn (2) is the node feature matrix. Thank you for the note and we have patched the notation explanation.
>
> - Similarly, we have included the domains of the matrices.

---

### Official Review · AnonReviewer1 · 2019-10-23
**Official Blind Review #1**

**Rating:** 6

**Review:**


The paper proposes a differentiable coarsening approach for graph neural network (GNNs).
To this end, it is motivated by algebraic multigrid and optimal transport methods.

GNNs is indeed an interesting line of research. And introducing coarsening into them, it a highly relevant step. However, there are some major downsides. First, some of the statements are a little but too strong. The paper starts with claiming that GNNs are competitive to graph kernels. But then for instance

Christopher Morris, Martin Ritzert, Matthias Fey, William L. Hamilton, Jan Eric Lenssen, Gaurav Rattan, Martin Grohe:
Weisfeiler and Leman Go Neural: Higher-Order Graph Neural Networks. AAAI 2019: 4602-4609

show that many (if not all) GNNs are equivalently expressive as the Weisefeiler-Lehman (WL) graph kernel. Hence, the competitiveness has to be qualified. Moreover, since you also employ graph convolutional networks for coarsening, you are also in the regime of this paper. Consequently, one should actually compare to WL, at least one should mention this connection. Actually, given that the datasets are not that large, one should run some statistical significance test. Moreover, if you check the paper above, they report much better results for PatchySan on MUTAG, better results on Protein for graph kernels, better results on IMDB-B using a hierarchical GNN approach, based on ideas of higher-order WL.

Nevertheless, indeed, the present paper shows that a differentiable pooling using WL kind of ideas is competitive to existing pooling approach. This is nice, but in the light of the work above, the novelty is unclear. This has to be clarified before publication.

**Experience Assessment:**

I have published in this field for several years.

**Review Assessment: Checking Correctness Of Derivations And Theory:**

I assessed the sensibility of the derivations and theory.

**Review Assessment: Checking Correctness Of Experiments:**

I carefully checked the experiments.

**Review Assessment: Thoroughness In Paper Reading:**

I read the paper thoroughly.

---

> ### Author Response · Authors · 2019-11-13
> **Response to Official Blind Review #1**
>
> Thank you very much for the considerate comments. The critique is largely concerned with WL. We respond to it from two angles and have updated the paper to incorporate these discussions.
>
> RE: GNN versus graph kernels.
>
> We well agree that there is a strong connection between WL kernels and various GNNs that fall under the “message passing” umbrella. And by no means we attempt to claim which one outperforms the other. From past empirical evidence, it is fair to say that results are often data set dependent (not to mention the overwhelming hyperparameter tuning cost). We would like to be open minded and not jump into hasty conclusions, in light of the current active development of GNNs, as well as efforts of identifying their expressive power.
>
> On the other hand, we would also like to stress that exploiting hierarchical structures, a focus of this paper, is an orthogonal effort to the WL connection. In fact, it would be an interesting topic to investigate whether one can inject coarsening into the WL test, as an attempt to improve computational efficiency in (approximate) graph isomorphism tests.
>
> We thank you for bringing up this issue. We have cited the mentioned paper, as well as related ones, and included a short discussion in the first paragraph of the introduction section.
>
> RE: Novelty.
>
> We would like to clarify that the coarsening approach we are proposing has a very weak connection with WL. The connection may be argued from the fact that we use GCN as one of the components in the parameterization of the coarsening matrix. However, the parameterization is only one piece of the method. The novelty and contribution of the work lie in the hierarchical treatment and the use of graph distance for parameter learning. The WL tests do not generate coarse graphs. The WL kernels also generally do not have a hierarchical flavor. Moreover, the counterpart of “graph distance” in the WL setting is the reproducing kernel Hilbert space, which is in contrast to optimal transport in our case. Hence, we debate that the proposed differentiable pooling does not “use WL kind of ideas” in a large part.
>
> To clarify the novelty and contribution, we have added a summary from the perspectives of unsupervised learning, coarsening strategy, and empirical results, at the end of the introduction section.

---

### Official Review · AnonReviewer3 · 2019-10-29
**Official Blind Review #3**

**Rating:** 6

**Review:**

This paper proposes an unsupervised hierarchical approach for learning graph representations. The proposed architecture is constructed by unrolling k-steps of a parametrized algebraic multigrid approach for minimizing the Wasserstein metric between the graph and its representation. The node distance (transport cost) used in the Wasserstein metric is also learned as an L2 distance between the embeddings of some graph embedding function. The approach is compared against 6 other state of the art approaches on 5 graph classification tasks, showing significant improvements 4 of them.

The paper is reasonably well written, however, I think some of the explanations can be tightened further. Especially a lot on the background of AMG is not really that relevant, since the authors are not transferring technical results from AMG. Also, it seems like a better flow for presenting this argument might be to switch the order of sections 3.2.1 and 3.1.2. It looks like the main point is  that this architecture is trying to emulate iterative coarsened residual optimization of the Wasserstein metric between a graph and its representation. How the coarsening matrix is derived is more of a technical point (it looks like the results would be much more sensitive to a switch of metric than to a switch of parametrization for S).

The empirical results are quite intriguing. There are, however, natural and important questions left unanswered. First and foremost, how does the amount of downsampling (compression) compare between methods. How many parameters do different methods require? It would also be good to see what the baseline performance would have been without any input compression as to understand how close these approaches are to the upper bound.

Finally, I think the main issue of this paper, is left unresolved, namely, what is the point of not having supervision from the downstream task. As a user of graph representations trying to solve some problem, the only thing I would want from my representation is to capture some notion of sufficient statistics that are small enough to be efficient and allow me to solve my problem. I would not necessarily care about how well the learned representation resembles the original graph unless I believed that my downstream task was hard to evaluate  and that it was very smooth in the Wasserstein metric. I read the paper multiple times, trying to find any discussion on this, but it seems that the fact that an unsupervised representation is a good thing is taken for granted. A point could at least be made using the same representation for different tasks experimentally. Or, perhaps, literally doing an AMG-type unpacking of the downstream task itself as a comparison. This would shed light on the question of whether the iterated residuals or the choice of distance is what's driving the observed results.

**Experience Assessment:**

I have read many papers in this area.

**Review Assessment: Checking Correctness Of Derivations And Theory:**

I assessed the sensibility of the derivations and theory.

**Review Assessment: Checking Correctness Of Experiments:**

I assessed the sensibility of the experiments.

**Review Assessment: Thoroughness In Paper Reading:**

I read the paper thoroughly.

---

> ### Author Response · Authors · 2019-11-13
> **Response to Official Blind Review #3**
>
> Thank you very much for the informative comments. In what follows we respond to them. We also have updated the paper accordingly.
>
> RE: Structure of the paper.
>
> We concur to your dissection of the work regarding the coarsening matrix and the Wasserstein metric. Indeed, we are not transferring results from AMG; but rather, we use it as an intuition of the design of the coarsening procedure A_c = S’AS. To put this component in context, we note that the design of coarsening (or “pooling”, used interchangeably) is a focus in the GNN literature recently. With flourishing proposals of the pooling operator, we do find that our proposal offers an opportunity to obtain meaningful coarse graphs, as demonstrated by the examples in Figure 3. We hope such a contribution will not be eclipsed by the use of Wasserstein metric to learn the coarsening matrix S.
>
> RE: Empirical results.
>
> The information regarding the amount of downsampling is presented in Appendix D, “Experimentation Details.” Specifically, the coarsening ratio is set to 0.5 for all methods, which means that every time the node count of the graph is halved. The number of levels, on the other hand, is treated a tunable hyperparameter. In most cases two levels lead to good results. In summary, the amount of compression is quite comparable.
>
> The number of parameters is also comparable for different hierarchical methods, since it is dominated by the GCN parameters. It, however, is roughly proportional to the number of coarsening levels, which, as mentioned, is a hyperparameter and is often set to two after tuning.
>
> We also updated the paper with baseline performance by ablating the coarsening. To make unsupervised training possible, in this case we use a graph autoencoder to obtain node representations and then pool them to form the graph representation. (The autocoder applies GCN twice, one to encode node features into the needed representations and the other to decode them and reconstruct the original node features.) The results are presented in the last but one row of Table 1. Overall, the baseline results are inferior to the proposed hierarchical method, but still are quite good in context. For more information, please see Sections 4.1 and 4.2.
>
> RE: The point of unsupervised learning.
>
> The major reason for conducting unsupervised learning is that in practice, labels are scarce and expensive to obtain. We are fortunate to have quite a few annotated data sets as benchmarks that facilitate evaluation. In real-life applications, however, often what limits the choice of methods is not the data but the labels (a folklore wisdom is that data is abundant in this “big data” era but labels are expensive to obtain).
>
> Additionally, we demonstrate the possibility of learning graph hierarchical structures without using labels.
>
> We concur that multitask learning is an excellent example demonstrating the use of a single representation for different downstream tasks. We have included an experiment and updated the paper; see Section 4.4.
>
> We also inserted the justification of unsupervised learning in the abstract.

---

### Public Comment · ~Mingxing_Xu3 · 2019-11-07
**Concerns about the insights in adopting optimal transport distance for unsupervised graph representation learning.**

It is a interesting work. This paper proposed a hierarchical unsupervised pooling operation.  In each pooling,  nodes are selected by keeping the top-k nodes after transforming the node feature to importance score with one-layer GCN to consider both node features and graph structures.  Then coarsening matrix $S$ are obtained by sampling and reweighing the normalized adjacent matrix. All the above strategy is existed. The main contribution of this paper is to adopt the optimal transport distance as unsupervised loss. In this paper, the optimal transportation distance are defined as $W_\gamma(G,G_c)=min_{P\in U(a,b)}<P,M>+\lambdaE(P)$. In this function, $M$ is transport matrix and obtained by calculating the distance between the  transformed node features of original graph and coarsening graph, $P$ is joint probability measure related to $M$ and the optimal joint probability measure can be obtained as $P_\lambda=diag(u)exp(-M/\lambda)diag(v)$. Thus the optimal transport distance is finally obtained and used to guide the training.
I have some concerns about the motivation. First,  why optimal transport distance is a good loss function to guide the selection of node and coarsening of graph, the insights behind are not fully discussed. Second, in final, we could obtain the transport cost matrix as well as the optimal joint probability measure. Does it mean that we can obtain the optimal transport plan to transport original graph to coarsening graph, what is the relationship between the optimal plan with the  coarsening matrix? this is not mentioned in this paper.

---

> ### Comment · AnonReviewer1 · 2019-11-12
> **A point in favour of the paper**
>
> Yes, I somehow agree that the technical part of the paper is not the main contribution. For me, it is the empirical demonstration that a differentiable pooling using WL kind of ideas is competitive to existing pooling approach. This is an interesting take-away message for me. But indeed the novelty is somewhat unclear. This has to be clarified before publication.

---

> ### Author Response · Authors · 2019-11-13
> **RE: Optimal transport**
>
> Thank you very much for raising the concerns. They are good questions. Let us respond from two angles.
>
> We choose to use optimal transport in a large part because it gives a convenient measure of the difference of two graphs. Recent work cited by the paper, such as Vayer et al. (2019), Xu et al., (2019a), and Garg & Jaakkola (2019), all devotes to the development of this measure. Natural alternatives may be to leverage graph embedding vectors and use the vector Euclidean distance as the measure, but we find that leveraging the node embedding vectors and treating them as a distribution in the optimal transport way is a more powerful machinery. Of course, one may summarize node embedding vectors into a graph embedding vector, but the summarization may lose information.
>
> The optimal transport plan P is obtained as a byproduct of the computation of the optimal transport distance. Conceptually, the plan may be interpreted as how much portion of what nodes are transported to a coarse node. On the other hand, the coarsening matrix S plays more a role of determining the edge weights of the coarse graph (because A_c = S’AS). The matrices P and S are not necessarily the same, not even the sparsity structure. We have thought of using P to replace S, but this is a chicken-and-egg problem, because parameterization cannot be done. Another challenge is that when one looks at P and S, one implicitly assumes that the coarse nodes are known. However, the selection of coarse nodes comes from the coarsening step but not the transportation step. One must resolve these technical challenges before being able to equate P with S.

---

### Author Response · Authors · 2019-11-13
**Summary of updates in the paper**

- We inserted a sentence in the abstract to justify unsupervised learning.
- We drew the connection to WL test and WL kernels in the first paragraph of the introduction section.
- We added a summary of the novelty and contributions at the end of the introduction section.
- We patched the meaning of X and dimensions of matrices in section 3.1.4.
- We included one additional data set, IMDB-MULTI, for graph classification.
- We included an unsupervised baseline for graph classification. See Sections 4.1 and 4.2 and Table 1.
- We expanded the experiments with multi-task learning. See Section 4.4.

---

### Decision · Program_Chairs · 2019-12-19

**Decision:**

Reject

**Comment:**

This paper presents a differentiable coarsening approach for graph neural network. It provides the empirical demonstration that the proposed approach is competitive to existing pooling approaches. However, although the paper shows an interesting observation, there are remaining novelty as well as clarity concerns. In particular, the contribution of the proposed work over the graph kernels based on other forms of coarsening such as the early work of Shervashidze et al. as well as higher-order WL (pointed out by Reviewer1) remains unclear. We believe the paper currently lacks comparisons and discussions, and will benefit from additional rounds of future revisions.